# Hydrolysis Behavior and Kinetics of AlN in Aluminum Dross during the Hydrometallurgical Process

**DOI:** 10.3390/ma15165499

**Published:** 2022-08-10

**Authors:** Hong-Liang Yang, Zi-Shen Li, You-Dong Ding, Qi-Qi Ge, Lan Jiang

**Affiliations:** 1School of Metallurgy, Northeastern University, Shenyang 110819, China; 2Key Laboratory for Ecological Metallurgy of Multimetallic Mineral, Ministry of Education, Northeastern University, Shenyang 110819, China

**Keywords:** solid waste, aluminum dross, AlN hydrolysis, kinetics

## Abstract

In this study, the hydrolysis behavior and kinetics of AlN in aluminum dross (AD) were investigated in order to better identify the steps controlling the AlN hydrolysis reaction and the factors influencing the hydrolysis rate to enhance the removal efficiency of AlN. The hydrolysis behavior of AlN, including AlN content, phase composition, chemical composition, microstructure, and element distribution, was determined by a leaching test, X-ray diffraction, X-ray fluorescence, scanning electron microscopy, and energy dispersive spectroscopy, respectively. The results showed that increasing the leaching liquid–solid ratio as well as the temperature was helpful for the removal efficiency of AlN. When the liquid-–solid ratio was 4:1, temperature was 90 °C, and leaching time was 300 min, the removal efficiency of AlN reached 89.05%. The kinetics were described using the unreacted core model, and when the temperature was 30–40, 50–70, and 80–90 °C, the hydrolysis reaction rate of AlN was controlled by boundary layer diffusion, chemical reaction control, and product layer diffusion control, respectively.

## 1. Introduction

Aluminum dross (AD) is produced in the process of Al metal smelting, Al alloy casting, and Al waste recycling [1,2]. It is a hazardous waste mainly consisting of Al_2_O_3_ and AlN [3,4,5,6]. The AlN in AD typically exhibits high reactivity [7], and it can react with water even at room temperature [8,9], yielding NH_3_, a toxic compound that has a risk of explosion when enriched to a certain concentration in the air. In addition, if the AD accumulates on wet ground, the NH_3_ will seep into the soil and destroy the pH of the soil, causing environmental pollution [10].

The effective removal of AlN in AD is important for environmental protection. Hydrometallurgical processing can effectively remove AlN from AD [11,12], and many studies have been performed in this research field. However, many factors affect the removal efficiency of AlN in AD, such as temperature [13,14,15,16,17,18], time [13,16,18], particle size [13,17], stirring speed [13,14], pH [13,18], liquid–solid ratio [13,16], additives [16,19], and pressure [20]. Studying the affecting factors contributes to kinetics analysis. Jiang et al. [13] reported that AlN in AD mostly exhibited small particles, which hardly hydrolyzed at 298 K, even when increasing the stirring rate or prolonging the time. When observing hydrolysis at 373 K for 24 h, the removal efficiency of AlN reached about 98%, indicating that higher temperatures and longer reaction times can increase the removal efficiency of AlN, while changing the stirring rate exhibited no obvious effect on the removal efficiency of AlN [21,22,23]. Lv et al. [24] reported that, based on orthogonal experiments, the factors affecting the removal efficiency of AlN were hydrolysis time, hydrolysis temperature, and stirring speed when the temperature was 100 °C. Han et al. [16] reported that increasing the liquid–solid ratio improved the removal efficiency of AlN when the temperature was 90 °C, while adding additives, such as AlF_3_, NaF, Na_2_CO_3_, and NaOH, also enhanced the removal efficiency of AlN. At 6 wt % NaOH, the removal efficiency of AlN reached about 93%. Qi et al. [18] reported that the hydrolysis reaction of AlN in AD was very slow when the temperature was lower than 50 °C. The surface of AlN particles was covered with a thin hydrolyzed layer of amorphous aluminum hydroxide or low-crystalline boehmite [22,25], inhibiting the hydrolysis rate of AlN. The reaction rate rapidly increased with the temperature. However, even at 100 °C, the removal efficiency of AlN was only about 68%. Li et al. [17] studied the hydrometallurgical disposing strategy of AD with a particle size smaller than 2000 μm. AlN mainly existed in smaller particles, which is consistent with the findings of Jiang’s research [13]. When the temperature was 25 °C, the hydrolysis reaction rate of AlN was slow, and the hydrolytic products deposited on the surface of AD particles, inhibiting the continuation of the hydrolysis reaction. Increasing the pH can improve the removal efficiency of AlN.

Most of these studies mainly focused on the hydrolysis behavior of AD, and only a few studies focused on the kinetics of AlN hydrolysis and the formation of hydrolysis reaction products. Moreover, it can be seen from these research results that the hydrolysis reaction of AlN in aluminum ash is very slow at lower temperatures [13,14,18,26] and very rapid at higher temperatures [11,16,24]. Moreover, at different temperatures, the factors affecting the removal efficiency of AlN in AD are inconsistent, and it is very difficult to completely remove the AlN from AD under various conditions. Thus, it is necessary to study the kinetics to identify the steps controlling the hydrolysis process.

Currently, there are some studies in this field. Han et al. [16] reported that in the range of 30–100 °C, the reaction rate of AlN in AD was mainly controlled by internal diffusion in an aqueous solution. The hydrolysis behavior conformed to the unreacted core model. Adding NaOH dissolved the product layer deposited on the surface of AlN particles, and the reaction became chemically controlled. Mostafa et al. [27] reported that in the range of 20–75 °C, the dissolution reaction rate of AD in hydrochloric acid was controlled by internal diffusion, while the dissolution behavior conformed to Jander’s model. Qun et al. [11] reported that in the range of 25–75 °C, the reaction rate of AlN in a hydrochloric acid solution was mainly controlled by the surface chemical reaction, whereas the dissolution behavior conformed to the shrinking volume model. Li et al. [17] reported that in the range of 25–80 °C, the hydrolysis reaction rate of AlN in AD was controlled by diffusion in a NaOH solution with an initial pH of 13, and the hydrolysis behavior conformed to the Avrami–Erofeyev model. These kinetic studies of the hydrolysis of AlN in AD indicate that the controlling steps are not the same in different solvent systems and at different temperatures; therefore, the kinetics of the hydrolysis of AlN in AD remain unclear.

The removal of AlN in AD is a complex process. The final result of the actual production process often depends not on the thermodynamic conditions, but on the speed of the reaction, that is, on the kinetic conditions. Therefore, the main objective of the present research was to investigate the hydrolysis behavior and kinetics of AlN in AD to find the factors affecting the hydrolysis rate of AlN and the steps controlling the hydrolysis process, so as to take targeted measures to strengthen it. This has important practical significance for the hydrometallurgical treatment of AD and the enhanced removal of AlN.

## 2. Experimental Materials and Methods

### 2.1. Materials

The results of X-ray diffraction (XRD) analysis of AD (from Inner Mongolia Hengsheng Environmental Protection Technology Co., Ltd., Chifeng, China), shown in Figure 1, indicated that AD comprises five phases: Al_2_O_3_, AlN, MgAl_2_O_4_, NaCl, and KCl. Figure 2 shows the microstructure and element distribution of AD, which indicate that AD is a complex agglomerate. The NaCl crystals are embedded on the particles with Al_2_O_3_ and MgAl_2_O_4_ as the main body, and AlN is scattered all over the surface of the particles. The chemical composition of AD is shown in Table 1. The deionized water was homemade and used throughout all experiments.

### 2.2. Experimental Procedure

The schematic diagram of the experimental equipment is shown in Figure 3. The experimental steps were as follows: in every experiment, 50 g of AD was weighed and preheated to the temperature, put into a beaker with deionized water, and thermostated into an automatic temperature-controlled water bath, ±0.1 °C. The solution was heated to the required temperature before AD was added. The stirring rate was controlled at 120 rpm, and after the reaction reached the set time, the system was quickly filtered. The filter cake was washed three times with deionized water, placed at room temperature for 24 h, and dried to constant weight at 105 °C. Finally, we determined the AlN content of the AD.

### 2.3. Analytical and Characterization Methods

The XRD patterns of the AD and leached residues were analyzed using an X-ray diffractometer (D8 ADVANCE, Bruker AXS Co., Ltd., Karlsruhe, Germany) with Cu-K_α_ (40 kV, 80 mA), the scanning range was 10 ~ 90°, and the scanning speed was 10°/min. The XRF analysis of the AD was conducted using an X-ray fluorescence spectrometer (ZSX Primus ii, Neo-Confucianism Co., Ltd., Tokyo, Japan). The optical tube voltage was 60 kV, and the current was 150 mA. Scanning electron microscopy (SEM) and energy dispersive spectroscopy (EDS) were performed with a field emission scanning electron microscope (Quanta 250FEG, FEI Co., Ltd., Brno, Czech Republic). The acceleration voltage was 30 kV. The particle size distribution of the AD was determined using a laser particle size analyzer (Bettersize 2000, Bettersize Instruments Co., Ltd., Dandong, China). The content of AlN in the AD was measured by chemical analysis. The schematic diagram of the measurement steps is shown in Figure 4. The dry sample was put into a digestion tube in Step 1, and then 10 mL of 1:1 phosphorus-sulfur mixed acid was added. Then, the digestion tube was put into a graphite digestion instrument (SH220F) for 60 min. After digestion was complete, 40 mL of 40 wt % NaOH solution was added to the digestion tube, which was placed into the Kjeldahl apparatus (K9840) in Step 2 for distillation. To absorb the distillate, 25 mL of a 2 wt % boric acid solution was used; in Step 3, the absorption solution was titrated with a 0.1 mol/L HCl solution, and methyl red-bromocresol green was used as an indicator. All reagents were analytically pure and used without further purification. The same sample was independently measured three times to minimize errors.

The AlN weight percent in the AD was calculated as follows:(1)$$\omega \;\%=\frac{c\times (V_{1}-V_{0})\times 41}{m\times 1000}\times 100\%$$
where ω is the AlN content in the AD, %; c is the molar concentration of hydrochloric acid, mol/L; *V_0_* is the blank titration volume, mL; *V_1_* is the titration volume, mL; and *m* is the sample mass, g.

The AlN removal efficiency was calculated as follows:(2)$$\eta \;\%=\frac{m_{1}\omega_{1}-m_{2}\omega_{2}}{m_{1}\omega_{1}}\times 100\%$$
where *η* is the AlN removal efficiency, %; *m*_1_ is the mass of the AD before hydrolysis, g; ω_1_ is the AlN content of the AD before hydrolysis, %; *m*_2_ is the mass of the AD after hydrolysis, g; ω_2_ is the AlN content of the AD after hydrolysis, %.

All data were independently measured three times to minimize measurement errors.

## 3. Results and Discussion

### 3.1. Effect of Different Factors on Removal Efficiency of SAD

#### 3.1.1. Effect of Liquid–Solid Ratio

Figure 5 shows the change in the removal efficiency of AlN in AD with increasing hydrolysis time at different liquid–solid ratios when the temperature was 90 °C. The results show that the removal efficiency of AlN changed the most in the first 60 min of the experiment, indicating that the hydrolysis of AlN mainly concentrated in the initial stage of the hydrolysis reaction. The increase in the liquid–solid ratio is beneficial to hydrolysis. The larger the liquid–solid ratio, the more complete the hydrolysis reaction of AlN, and the higher the removal efficiency of AlN. Due to the increase in the liquid–solid ratio, the slurry viscosity and the internal diffusion transfer resistance of the hydrolysis process decreases, and the hydrolysis reaction rate of AlN gradually increases. However, too large of a liquid–solid ratio does not significantly improve the removal efficiency of AlN. This is because increasing the liquid–solid ratio reduces the concentration of ammonia in the leachate, and the alkaline environment created by the dissolution of ammonia is conducive to the hydrolysis of AlN [16].

#### 3.1.2. Effect of Temperature

Figure 6 shows the change in the removal efficiency of AlN in AD with increasing hydrolysis time at different temperatures when the liquid–solid ratio was 4:1. The results show that increasing the temperature improved the removal efficiency and reaction rate of AlN. When the temperature was in the range of 30–40 °C, the removal efficiency of AlN slowly increased with time. Some earlier researchers also observed a similar slow reaction stage before a fast hydrolysis reaction [23,28]. This was attributed to a thin layer of amorphous AlOOH [25] wrapped on the surface of the AlN particles. While in contact with moisture, it acts as a barrier preventing further contact between AlN particles and water [17,18,22], making a protective film on the surface of solid particles and hindering the reaction progress. The amorphous AlOOH can be transformed into crystalline Al(OH)_3_ [22,23,25], a transformation process that accelerates with temperature [28]. Li et al. [29] showed that the presence of NaCl in the solution can enhance the solubility of aluminum hydroxide gel and accelerate the hydrolysis of AIN, resulting in a faster reaction. With the increase in temperature, after the slow reaction stage, the hydrolysis reaction rate of AlN accelerates. The higher the temperature, the shorter the time required to reach a plateau in the AlN removal efficiency. The thermodynamic calculation of AlN hydrolysis [30] shows that the formation of reaction products AlOOH and Al(OH)_3_ are both exothermic, with a ΔH of −105.4 and −81.2 kJ/mol, respectively. The heat released by this exothermic reaction can further promote the AlN hydrolysis reaction. Therefore, the higher the temperature, the greater the AlN removal efficiency. The removal efficiency of AlN reached 89.05% when the hydrolysis treatment was performed at 90 °C for 300 min, and the removal efficiency of AlN remained almost unchanged by continuing to prolong the treatment time.

### 3.2. Analysis of Hydrolysis Products

#### 3.2.1. Analysis of XRD

Figure 7 shows the XRD patterns of the AD after hydrolysis for 120 min at different temperatures, confirming that higher temperatures are beneficial to the hydrolysis of AlN. Figure 8 shows the phase change of the AlN hydrolysis reaction products at different temperatures for 120 min. The results in Figure 8 show that the final reaction products of AlN hydrolysis are mainly AlOOH and Al(OH)_3_. The removal efficiency of AlN reached 16.42% at 50 °C for 120 min (in Figure 6); however, there is no obvious diffraction peak of the reaction products (AlOOH or Al(OH)_3_) in Figure 7a, which may be due to the relatively low content or the low crystallinity that made it difficult to identify them from faint reflections. When the temperature was raised to 60 °C, as shown in Figure 7b, there was a diffraction peak of the AlOOH phase but no diffraction peak of the Al(OH)_3_ phase. When the temperature was 90 °C, as shown in Figure 7c, the diffraction peaks of the AlOOH and Al(OH)_3_ phases appeared in the reaction product of AlN. This is consistent with the research conclusion of Bowen et al. [31]; the reaction of AlN hydrolysis in ionized water is shown in Equation (3):AlN + 2H_2_O→(AlOOH)_amorphous_ + NH_3_
(3)

NH_3_ is ionized in water, and the pH of the solution increases according to Equation (4). In an alkaline environment, amorphous AlOOH transforms into stable crystalline Al(OH)_3_, according to Equation (5), and this transition behavior more readily occurs at higher temperatures. Figure 8 shows the phase change of AlN in AD after hydrolysis for different times when the temperature was 90 °C, and the diffraction peaks of the AlOOH phase appeared when the hydrolysis treatment was 20 min, which further illustrates the AlN in AD. The hydrolysis products of AlN differ with the different hydrolysis temperatures: the hydrolyzed product of AlN is more likely to form AlOOH first when the temperature is higher than 351 K [28,30]. With the prolongation of the treatment time, it gradually transforms into Al(OH)_3_. The results in Figure 8c,d also show this.
(4)$${\mathrm{NH}}_{3}+H_{2}O↔{\mathrm{NH}}_{4}^{+}+{\mathrm{OH}}^{-}$$
$$\left({\mathrm{NH}}_{3}+H^{+}↔{\mathrm{NH}}_{4}^{+}\right)$$
(5)$${\left(\mathrm{AlOOH}\right)}_{\mathrm{amorphous}}+H_{2}O\to \mathrm{Al}{\left(\mathrm{OH}\right)}_{3\mathrm{crystalline}}$$

#### 3.2.2. Analysis of SEM and EDS

Figure 9 shows the SEM images of AD after hydrolysis for 120 min under different temperature conditions when the liquid–solid ratio was 4:1. Figure 9a,c,e shows the microstructures of the particles when the temperature was 50, 60, and 90 °C, respectively, and Figure 9b,d,f shows the corresponding enlarged image of the local area. It can be seen from Figure 9a,b that when the temperature was 50 °C, the hydrolysis product was boehmite(AlOOH) with a villi-like structure [23], which was evenly wrapped on the surface of each particle. When the temperature reached 60 °C, the hydrolysis product not only contained boehmite, but also a small amount of rod-shaped particles, and further transformed from boehmite with lower crystallinity to large cone- or rod-shaped bayerite [Al(OH)_3_], as shown in Figure 9c,d. This occurred due to the rapid breaking of Al-O-Al and Al-OH-Al bonds of thin-skinned diaspore (AlOOH) that then formed in a well-crystallized bayerite phase in the alkaline solution. When the temperature was 90 °C, as shown in Figure 9e,f, the rod-shaped particles significantly increased, and the formation of the hydrolyzed product AlOOH simultaneously occurred with the crystallization of Al(OH)_3_. It is reasoned that the hydrolysis products tend to nucleate and grow around the parent particles, subsequently forming agglomerates, which appear as bunches of rods. Based on the above analysis, we suggest that dissolution and recrystallisation mechanisms are involved in the transformation of AlN in AD from boehmite to bayerite.

Figure 10 shows the microstructure and element distribution of AD after hydrolysis treatment for 300 min at a liquid–solid ratio of 4:1 and a temperature of 90 °C. It can clearly be seen from the comparison with the untreated AD (Figure 2) that the element distribution in the treated AD was mainly Al, O, and Mg, and almost no N, Na, or Cl, which is consistent with the XRD analysis results in Figure 8d, indicating that AlN and NaCl in AD can be effectively removed under these conditions.

### 3.3. Kinetic Analysis

The hydrolysis process of AlN in AD is a liquid–solid phase heterogeneous reaction at the phase interface. The unreacted core model is widely used to evaluate liquid–solid reactions. For an irreversible reaction, there are three regimes of control [22,30]: (i) mass transfer of reactant through a film surrounding the particle to the surface of the solid, (ii) diffusion of the reactant through the product layer to the surface of the unreacted core, and (iii) chemical reaction of reactant and solid at the core surface. The slowest step is rate-controlling. For the hydrolysis reaction of AlN in AD, the control steps are not static. It is possible that the reaction rate constants of these three control mechanisms are similar, and two or three links simultaneously control the rate of the whole process, which is the so-called “mixed kinetics” control process [32]. The equation of the hydrolysis reaction is represented by a general formula, as shown in Equation (6). From the analysis of Figure 7, Figure 8, Figure 9 and Figure 10, it is seen that during the hydrolysis of AlN, there is an insoluble solid product layer, the reaction area of the particles may change with time, and the hydrolysis reaction process undergoes multiple steps. The general formula of the kinetic equation is derived after treatment with the steady-state approximation as Equation (7):(6)$$aA_{\left(s\right)}+bB_{\left(g,l\right)}=pP_{\left(s\right)}+qQ_{\left(g,l\right)}$$
(7)$$\frac{b\delta }{D_{s}}\eta +\frac{3br_{0}}{2D'}\left[1-\frac{2}{3}\eta -{\left(1-\eta \right)}^{\frac{2}{3}}\right]+\frac{1}{k_{0}}\left[1-{\left(1-\eta \right)}^{\frac{1}{3}}\right]=\frac{3cM}{\rho r_{0}}t$$
where *δ* is the thickness of the boundary layer; *D_s_* is the diffusion coefficient through the liquid phase; *D’* is the diffusion coefficient through the solid product layer; *K_0_* is the surface chemical reaction rate constant; *b* is the stoichiometric coefficient of B in the reaction formula; *c* is the bulk concentration of reactant B in the solution; *ρ* is the density of reactant A; *M* is the relative molecular mass or relative atomic mass of reactant A; r_0_ is the initial radius of the particle.

According to the experimental data in Figure 8, the analysis results of the relationship between different kinetic equations and times are shown in Figure 11. The regression coefficient (R^2^) indicates the degree of agreement between the experimental data and the kinetic model. If the mathematical equations, with respect to experimental data (η_AlN_), are plotted versus time, the optimal reaction model should give a straight line.

Combined with the previous analysis, the results show that the control step of the AlN hydrolysis reaction in AD is affected by the temperature, which cannot be analyzed by a single kinetic equation in the range of 30–90 °C. When the temperature is low, the removal efficiency of AlN is very low, and it is in the slow reaction stage that the reactants transfer mass to the solid surface through the membrane surrounding the particles. At this time, $\frac{b\delta }{D_{s}}\;\gg$ $\frac{3br_{0}}{2D'}$ and $\frac{b\delta }{D_{s}}\;\gg$ $\frac{1}{k_{0}}$, so the second and third items on the left side in Equation (7) can be ignored. The kinetic equation is expressed as Equation (8), $k_{1}=\frac{3D_{S}cM}{b\delta \rho r_{0}}$, and the analysis result is shown in Figure 11a. When the temperature is in the range of 30–50 °C, the removal efficiency of AlN is more consistent with Equation (8), and the hydrolysis reaction rate of AlN is mainly controlled by boundary layer diffusion.
(8)$$k_{1}t=\eta$$

With the increase in temperature, the gel layer on the surface of AlN particles dissolves, and the hydrolysis reaction rate gradually increases. At this time, $\frac{1}{k_{0}}$ ≫ $\frac{b\delta }{D_{s}}$ and $\frac{1}{k_{0}}$ ≫ $\frac{3br_{0}}{2D'}$, so the kinetic equation is expressed as Equation (9), $k_{2}=\frac{3k_{0}cM}{\rho r_{0}}$. The analysis results are shown in Figure 11b. In the range of 30–50 °C, the removal efficiency of AlN is more consistent with Equation (9), and the hydrolysis reaction rate of AlN is mainly controlled by the surface chemical reaction.
(9)$$\;k_{2}t=1-{\left(1-\eta \;\right)}^{1/3}$$

When the temperature is in the range of 70–90 °C, the relationship between the removal efficiency of AlN and time can be described by the parabolic law, as shown in Figure 6. The hydrolysis reaction product generates a dense solid product layer and accumulates on the particle surface (Figure 9), and the diffusion resistance to the reaction product is much larger than the external diffusion. At this time, $\frac{3br_{0}}{2D'}$ ≫ $\frac{b\delta }{D_{s}}$ and $\frac{3br_{0}}{2D'}$ ≫ $\frac{1}{k_{0}}$, so the kinetic equation is expressed as Equation (10), $k_{3}=\frac{2D^{\prime }cM}{b\rho r_{0}^{2}}$, and the analysis result is shown in Figure 11c. The hydrolysis reaction rate of AlN is mainly controlled by the diffusion of the solid product layer.
(10)$$k_{3}t=1-\frac{2}{3}\eta -{\left(1-\eta \right)}^{2/3}$$

At the temperatures of 50 and 70 °C, the hydrolysis reaction rate of AlN may be controlled by the mixing steps, which are boundary layer diffusion and surface chemical reaction, and surface chemical reaction and solid product layer diffusion, respectively. Therefore, under the conditions of different temperatures, we can obtain the factors affecting the hydrolysis reaction rate of aluminum nitride according to $k_{1}$, $k_{2}$, and $k_{3}$, which provides effective theoretical support for strengthening the removal efficiency of AlN.

The Arrhenius equation shows the apparent rate constant of the chemical reaction, k, versus temperature T (K), so we can calculate the apparent activation energy (Ea) of the AlN hydrolysis reaction from Equation (11).
(11)$$\mathrm{lnk}=\mathrm{lnA}-\frac{E_{a}}{\mathrm{RT}}$$
where k is the rate constant, R is the molar gas constant, T is the thermodynamic temperature, Ea is the apparent activation energy, and A is the pre-exponential factor.

The results are shown in Table 2. The calculation results show that increasing the temperature of the reaction can effectively reduce the apparent activation energy of the AlN hydrolysis reaction, thereby making the hydrolysis reaction easier.

## 4. Conclusions

In this study, we discussed the effect of different liquid–solid ratios and temperatures on the removal efficiency of AlN and the kinetics of AlN hydrolysis in AD. The main findings can be summarized as follows:

In the process of hydrometallurgical treatment of AD with deionized water as the solution, increasing the liquid–solid ratio as well as the temperature can increase the removal efficiency of AlN and accelerate the hydrolysis reaction rate. The removal efficiency of AlN reached 89.05% at the optimum hydrolysis conditions (leaching time of 300 min, temperature of 90 °C, and liquid–solid ratio of 4:1). The hydrolysis behavior of AlN in AD is a complex process, and the hydrolysis products are mainly AlOOH and Al(OH)_3_. The kinetic analysis of the hydrolysis behavior of AlN in AD in deionized water conforms to the unreacted core model. When the temperature is 30–40, 50–70, and 80–90 °C, the hydrolysis reaction rate of AlN is mainly controlled by boundary layer diffusion, chemical reaction control, and product layer diffusion control, respectively. The apparent activation energy of the AlN hydrolysis reaction is 96.53, 87.13, and 34.59/mol, respectively. Increasing the temperature can effectively reduce the apparent activation energy of the AlN hydrolysis reaction, thereby making the hydrolysis reaction easier.

## Figures and Tables

**Figure 1 materials-15-05499-f001:**
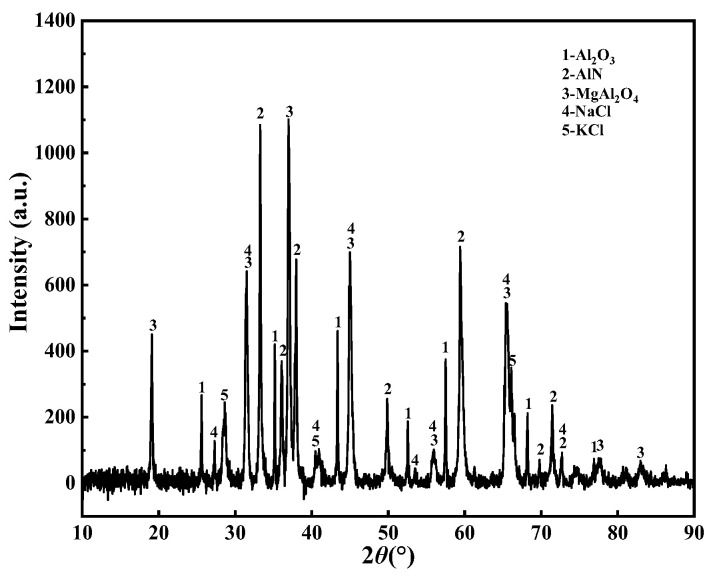
XRD patterns of AD.

**Figure 2 materials-15-05499-f002:**
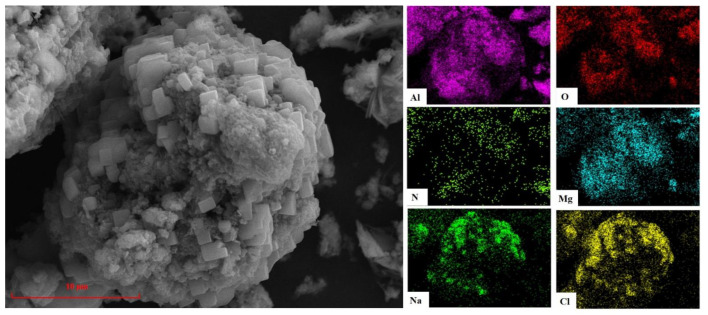
SEM and EDS of AD.

**Figure 3 materials-15-05499-f003:**
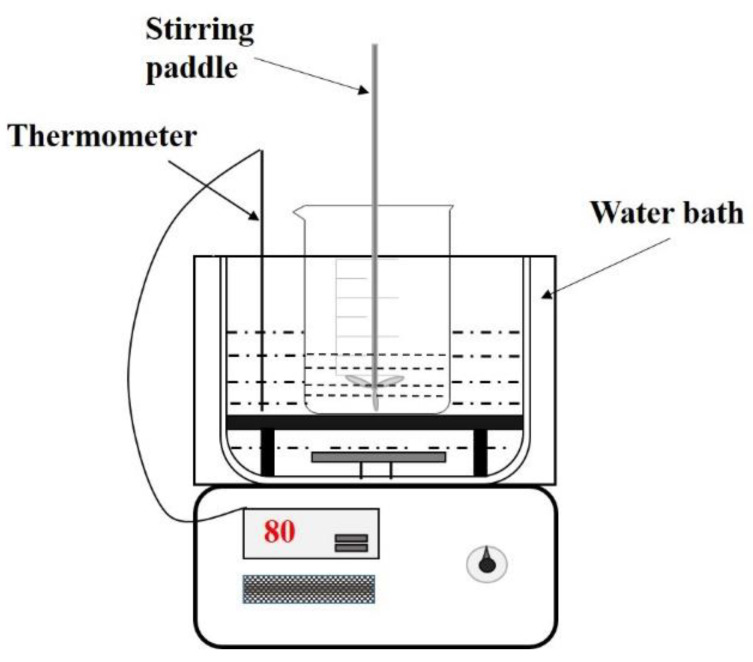
Schematic diagram of the experimental setup.

**Figure 4 materials-15-05499-f004:**
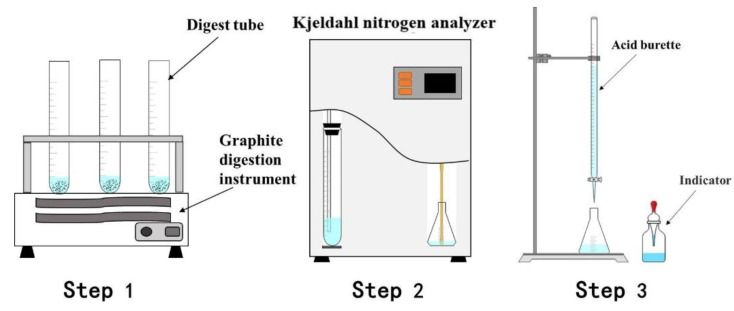
Schematic diagram of the measurement steps used to determine the AlN content.

**Figure 5 materials-15-05499-f005:**
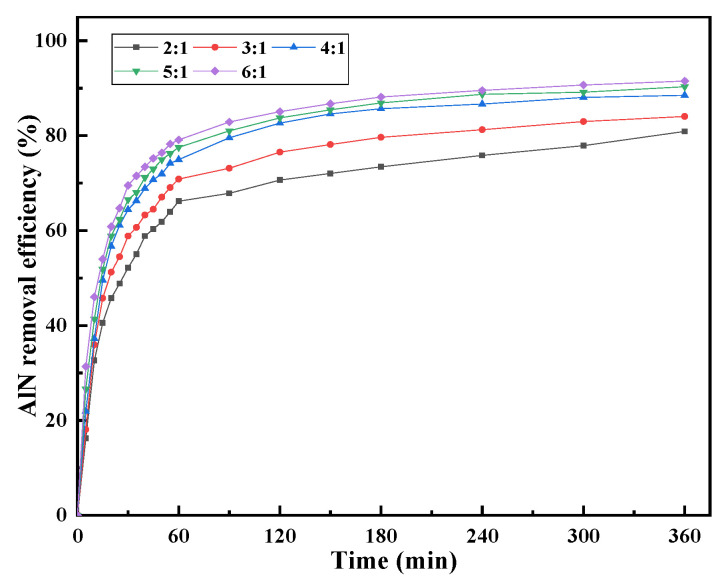
Effect of the liquid–solid ratio on the AlN removal efficiency at a temperature of 90 °C.

**Figure 6 materials-15-05499-f006:**
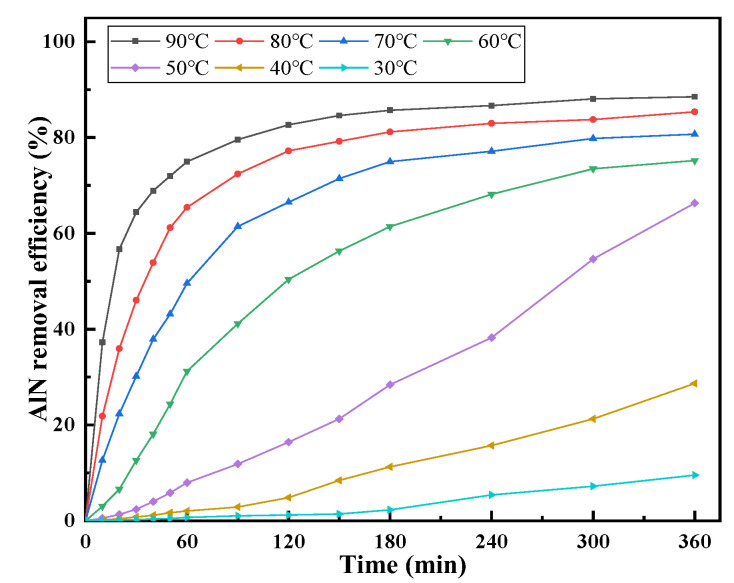
Effect of the temperature on the AlN removal efficiency at a liquid–solid ratio of 4:1.

**Figure 7 materials-15-05499-f007:**
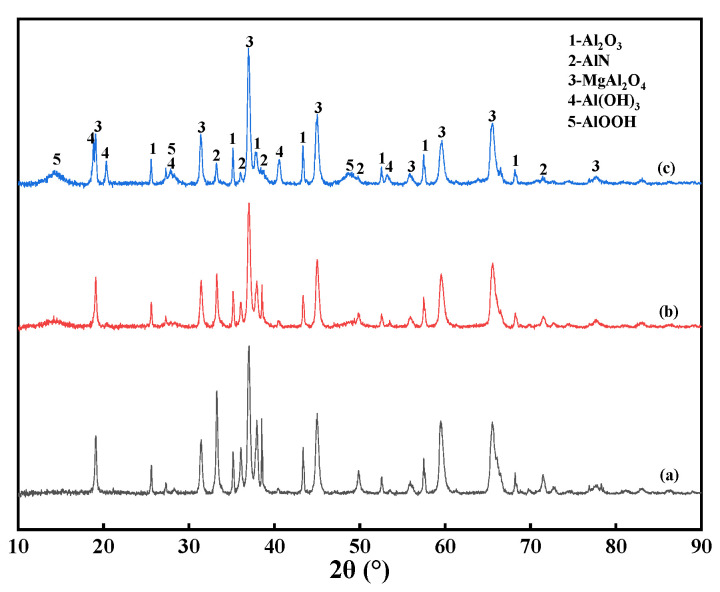
XRD patterns of AD after hydrolysis for 120 min at different temperatures. (**a**) 50 °C; (**b**) 60 °C; (**c**) 90 °C.

**Figure 8 materials-15-05499-f008:**
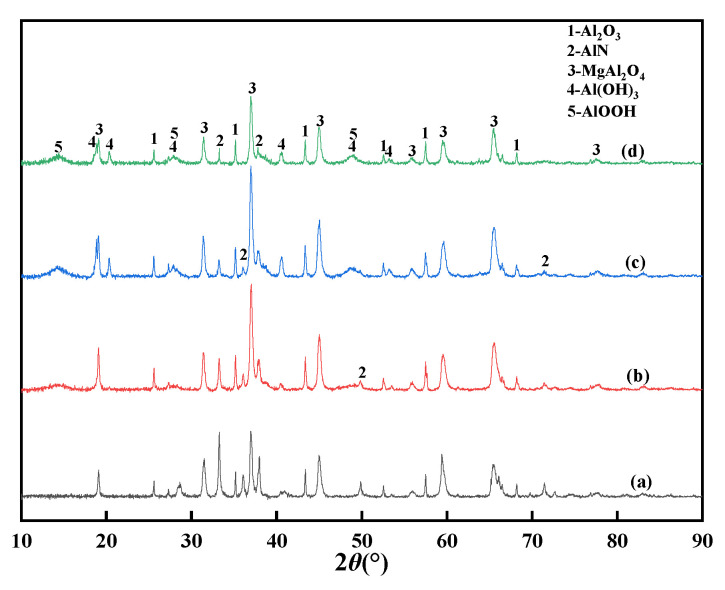
XRD patterns of AD after different treatment times at a temperature of 90 °C: (**a**) 0; (**b**) 20; (**c**) 120; (**d**) 300 min.

**Figure 9 materials-15-05499-f009:**
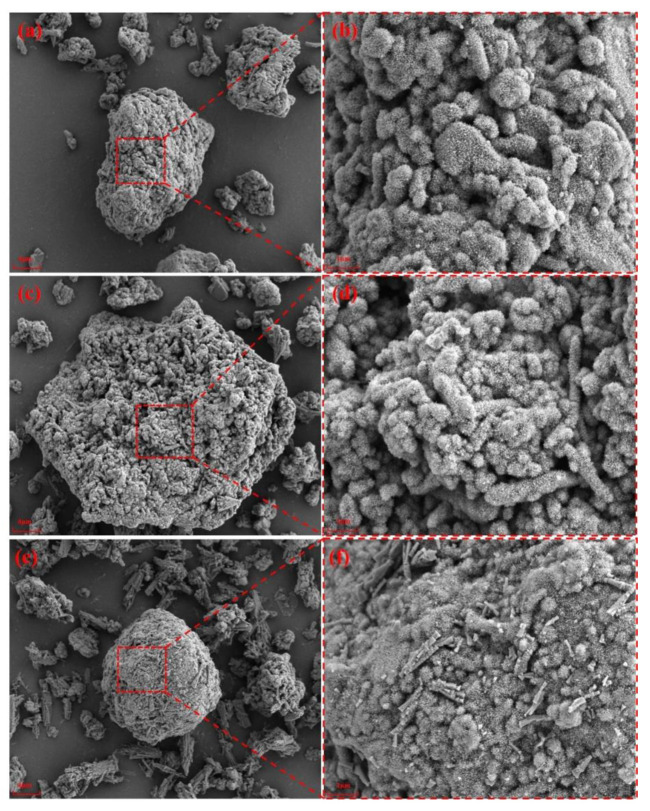
SEM images of AD hydrolyzed for 120 min under different temperature conditions with liquid–solid ratio of 4:1: (**a**,**b**) 50; (**c**,**d**) 60; (**e**,**f**) 90 °C.

**Figure 10 materials-15-05499-f010:**
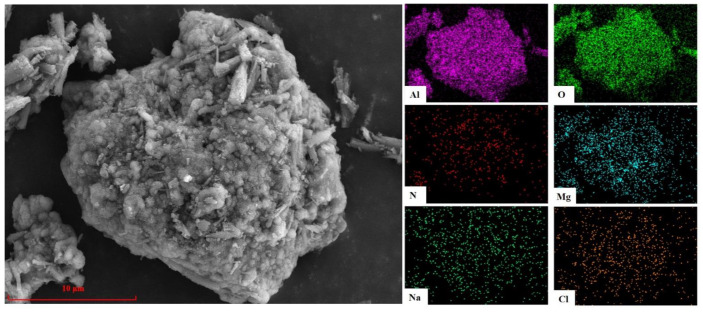
Microstructure and element distribution of AD after hydrolysis treatment for 300 min at liquid–solid ratio of 4:1 at 90 °C.

**Figure 11 materials-15-05499-f011:**
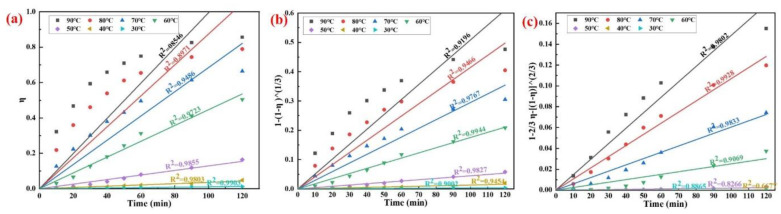
A representation of three various kinetic models fit to the experimental data. (**a**) η vs. time; (**b**)$1-\frac{2}{3}\eta -{\left(1-\eta \right)}^{2/3}$ vs. time; (**c**) $1-{\left(1-\eta \;\right)}^{1/3}$ vs time.

**Table 1 materials-15-05499-t001:** Chemical analysis of AD.

Element	O	Al	N	Mg	Na	K	Cl	Si	Ca	Ti	Else
**Content (%)**	29.56	47.13	9.49	4.78	2.21	1.09	4.40	0.43	0.31	0.18	0.42

**Table 2 materials-15-05499-t002:** Estimated rate-controlling steps with corresponding rate constants k and apparent activation energies Ea for the hydrolysis of AlN in AD in the range of 30–90 °C.

Temperature (°C)	Rate-Controlling Step	k	Ea (kJ/mol)
**30**	Boundary layer diffusion	1.05 × 10^−4^	96.53
**40**	Boundary layer diffusion	3.57 × 10^−4^	
**50**	Boundary layer diffusion and chemical reaction	4.49 × 10^−4^	87.13
**60**	Chemical reaction	1.76 × 10^−3^	
**70**	Chemical reaction and product layer diffusion	2.95 × 10^−3^	
**80**	Product layer diffusion	1.02 × 10^−3^	34.59
**90**	Product layer diffusion	1.42 × 10^−3^	

## Data Availability

The data presented in this study are available on request from the corresponding author. The data are not publicly available due to industrial application and confidentiality.

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
