# Peer review of "Hydrolysis Behavior and Kinetics of AlN in Aluminum Dross during the Hydrometallurgical Process"

_materials, 2022, doi:10.3390/ma15165499_

Round 1

Reviewer 1 Report

Comments to the authors:

0. In the introduciton, second paragraph, 6th line from above: 
Please remove the etc. or introduce some references for these etc?

1. What are the hazards of AlN if aluminum dross is dumped in the land field of the industry? Could the authors write some sentences about it in the first paragraph of the introduction?

2. How is the Table 1 provided? is it the results from XRF? If so, how do the auhtors see XRF technique accurate for measuring nitrogen.

3. If Table 1 is provided by XRF measurments, could the authors include the XRF analysis of the hydrolyzed samples after experiments? at leas one or two is highly suggested. This will help to show How the AlN content changes in the samples and can be compared with the weight loss data. 

4. What could be the draw back of this method for dissolving the AlN? as a point, did the authors detect PH3 (phosphine gas) formtion while doing this hydrolysis?

5. Figure 11 shall be presented in higher quality.

6. Caption of Figure 11 shall be cahnged to : A representation of three various kinetic models fit to the experimental data ......

Author Response

Thank you for your letter and for the referee’s comments concerning our manuscript entitled “Investigations on the hydrolysis behavior and kinetics of AlN in aluminum dross during the hydrometallurgical process”, manuscript number materials-1850379. We have checked the referee’s comments very carefully and modified the paper substantially according to the referee’s review. Revised portion have been highlighted in red with the original text.

0. In the introduciton, second paragraph, 6th line from above: 
Please remove the etc. or introduce some references for these etc?

Answer:Thanks for your suggestion, we've change remove the etc.

  1. What are the hazards of AlN if aluminum dross is dumped in the land field of the industry? Could the authors write some sentences about it in the first paragraph of the introduction?

Answer:Thanks for your suggestion, we've rewritten the first paragraph of the introduction.

2. How is the Table 1 provided? is it the results from XRF? If so, how do the auhtors see XRF technique accurate for measuring nitrogen.

Answer:Thanks a lot for your remark to this point. Due to the complexity and diversity of the AD, the composition analysis of AD relies on a variety of detection methods. First, XRF was used to obtain the types of elements contained in AD, and then chemical analysis methods are used to measure the element content in AD. Finally, the main phase types were identified by XRD, and the content of the main components of AD was listed in Table 1.

As for the XRF technique accurate for measuring nitrogen. Since the relative atomic mass of N is too small, it is not accurate to measure its content by XRF. In this paper, the N element in AD was obtained according to the chemical analysis method in Section 2.3. In order to avoid misunderstanding, we list the composition of AD in Table 1 in the form of elemental content and has been changed in the paper.

3. If Table 1 is provided by XRF measurments, could the authors include the XRF analysis of the hydrolyzed samples after experiments? at leas one or two is highly suggested. This will help to show How the AlN content changes in the samples and can be compared with the weight loss data. 

Answer:Thanks for your suggestion.

As mentioned above, Table 1 is not only provided by XRF. Since the investigation of the hydrolysis behavior and kinetics of AlN in AD was in order to better identify the controlling steps of the AlN hydrolysis reaction and the influencing factors of the hydrolysis rate. Therefore, we only discuss the variation of AlN content, and the accurate measurement method of AlN content is listed in Section 2.3. In addition, since the AD treated in deionized aqueous solution, not only the weight gain caused by the hydrolysis reaction of AlN, but also the weight loss caused by washing away the soluble salts, the determination of the AlN content in Section 2.3 comprehensively considers the weight change. That’s why we did not give the composition of the AD after the experiments, only show the changes of the AlN content in the samples.

4. What could be the draw back of this method for dissolving the AlN? as a point, did the authors detect PH3 (phosphine gas) formtion while doing this hydrolysis?

Answer:Thanks a lot for your remark to this point.

When deionized water is used as a solvent to hydrolyze AlN, the hydrolyzate gradually accumulates on the surface of AlN particles, wrapping the unreacted core and hindering the further hydrolysis reaction, making it difficult for the hydrolysis of AlN to reach a removal efficiency of 100%. However, the advantage of this method is that the transformation mechanism of AlN hydrolysis can be better analyzed, and it is convenient to identify the control steps of AlN hydrolysis reaction.

Furthermore, we did not detect PH3 (phosphine gas) formtion while doing this hydrolysis

5. Figure 11 shall be presented in higher quality.

Answer:As your suggested, we've change the Figure 11

6. Caption of Figure 11 shall be cahnged to : A representation of three various kinetic models fit to the experimental data ......

Answer:As your suggested, we've change the caption of Figure 11

Reviewer 2 Report

The work presented in this study describes the way in which the hydrolysis of aluminum nitrate formed on the surface of aluminum dross. The observations and explanation of the process is based on the common knowledge of the principles applicable to such processes. The part of the kinetic study is a little ambiguous and needs some statements describing the process mechanism. I recommend the paper be accepted for publication with minor revision.

3.1.2 Effect of initial temperature – The authors mention throughout the paper, the term “initial” but there is no behavior described justifies it. In fact, the term “initial “is meaningless in this study. Remove the term “initial” unless the behavior refers to true initial stage which is different from later stages.

Eq 3 is an abbreviated form and should be written in full.  

Eq 7 – reference should be given. Assumptions involved in the equation should be briefly mentioned. It appears to assume that diffusion is the rate limiting without full justification of the system.

The work presented in this study describes the way in which the hydrolysis of aluminum nitrate formed on the surface of aluminum dross. The observations and explanation of the process is based on the common knowledge of the principles applicable to such processes. The part of the kinetic study is a little ambiguous and needs some statements describing the process mechanism. I recommend the paper be accepted for publication with minor revision.

3.1.2 Effect of initial temperature – The authors mention throughout the paper, the term “initial” but there is no behavior described justifies it. In fact, the term “initial “is meaningless in this study. Remove the term “initial” unless the behavior refers to true initial stage which is different from later stages.

Eq 3 is an abbreviated form and should be written in full.  

Eq 7 – reference should be given. Assumptions involved in the equation should be briefly mentioned. It appears to assume that diffusion is the rate limiting without full justification of the system.

Author Response

Thank you for your letter and for the referee’s comments concerning our manuscript entitled “Investigations on the hydrolysis behavior and kinetics of AlN in aluminum dross during the hydrometallurgical process”, manuscript number materials-1850379. We have checked the referee’s comments very carefully and modified the paper substantially according to the referee’s review. Revised portion have been highlighted in red with the original text.

3.1.2 Effect of initial temperature – The authors mention throughout the paper, the term “initial” but there is no behavior described justifies it. In fact, the term “initial “is meaningless in this study. Remove the term “initial” unless the behavior refers to true initial stage which is different from later stages.

Response:As your suggested, we've remove the term “initial”

Eq 3 is an abbreviated form and should be written in full.

Response:As your suggested, we've change Eq to equation

Eq 7 – reference should be given. Assumptions involved in the equation should be briefly mentioned. It appears to assume that diffusion is the rate limiting without full justification of the system.

Response:Thanks a lot for your remark to this point.

We have given the corresponding references in the paper and briefly introduced the assumptions of the equation 7

Reviewer 3 Report

Abstract is excellently written. It consist of all parts that good Abstract should have.

Introduction structure is well formed. Follow up the comments provided bellow with more detailed analysis of this section:

Consider to change first sentence with two shorter, which will give a better provide into origin and content of mentioned AD.

In the whole Introduction, numbers of atoms that are part of molecule aluminum (III) oxide, ammonia and potassium carbonate should be transferred in subscript.

More, in the sentence: ”Moreover, it can be seen from these research results that The hydrolysis reaction of AlN in aluminum…”, you have a grammar error (big letter). Same error in the next sentence:” Moreover, At different temperatures, the factors…”.

Also, in the sentence: ”Increasing the pH can improve the the removal efficiency…”, you wrote article the two time.

Further, in the sentence: ”Thus, it is necessary to study the kinetics identify the control steps of the hydrolysis process.”, missing of verb before identify (properly would be to identify).

In the sentence:” Currently, there are some results in this field. Han Lv[16]…”, format used reference as in whole Paper. Do this in all other cases through manuscript.

Additionally, “Apparently, the kinetic studies of the hydrolysis of AlN in AD indicate that control steps are not the same in different solution systems…”, term solution could be change with solvent as a more precisely.

Highlight your innovations/advances and differences compared to available literature (why such approach could bring benefits after implementation).

Experimental:

Materials: Great representation of initial materials from this study. Highly recommend to put all used chemicals in this part of Paper, with their origins and purity.

Experimental procedures

Figures 3 and 4: Scheme of applied experimental equipment are nice represented.

Author Response

Thank you for your letter and for the referee’s comments concerning our manuscript entitled “Investigations on the hydrolysis behavior and kinetics of AlN in aluminum dross during the hydrometallurgical process”, manuscript number materials-1850379. We have checked the referee’s comments very carefully and modified the paper substantially according to the referee’s review. Revised portion have been highlighted in red with the original text.

Abstract is excellently written. It consist of all parts that good Abstract should have.

Response:Thank you very much for the encouragement and recognition for this work.

Introduction structure is well formed. Follow up the comments provided bellow with more detailed analysis of this section:

Response:Thank you very much for the encouragement and recognition for this work.

Consider to change first sentence with two shorter, which will give a better provide into origin and content of mentioned AD.

Response:Thanks for your suggestion, we've rewritten the first paragraph of the introduction

In the whole Introduction, numbers of atoms that are part of molecule aluminum (III) oxide, ammonia and potassium carbonate should be transferred in subscript.

Response:Thanks for your suggestion, we checked these errors carefully. All the numbers of atoms in the Word format are in subscript form, but when converted to the PDF format of the submitted version, the numbers of these chemical formulas are not subscripted, which also confuses us. Details can be found at see in the attachment

More, in the sentence: ”Moreover, it can be seen from these research results that The hydrolysis reaction of AlN in aluminum…”, you have a grammar error (big letter). Same error in the next sentence:” Moreover, At different temperatures, the factors…”

Response:As your suggested, we've change “The” to “the”, and “At” to “at”

Also, in the sentence: ”Increasing the pH can improve the the removal efficiency…”, you wrote article the two time.

Response:As your suggested, we've rewrite the sentence.

Further, in the sentence: ”Thus, it is necessary to study the kinetics identify the control steps of the hydrolysis process.”, missing of verb before identify (properly would be to identify).

Response:As your suggested, we've add the verb to before identify.

In the sentence:” Currently, there are some results in this field. Han Lv[16]…”, format used reference as in whole Paper. Do this in all other cases through manuscript.

Response:As your suggested, we've change the format used reference in whole paper.

Additionally, “Apparently, the kinetic studies of the hydrolysis of AlN in AD indicate that control steps are not the same in different solution systems…”, term solution could be change with solvent as a more precisely.

Response:As your suggested, we've change the term solution to solvent .

Highlight your innovations/advances and differences compared to available literature (why such approach could bring benefits after implementation).

Response:Thanks a lot for your remark to this point.

Compared to available literature, we used deionized water as a solvent to deal with the AD by hydrometallurgical process, and studied the hydrolysis behavior of AlN in AD. The transformation mechanism of AlN hydrolysis at different temperatures is more completely analyzed, and the control steps of AlN hydrolysis reaction under different temperature conditions are identified. Therefore, it can provide a convincing theoretical basis for the targeted strengthening of removal of AlN from AD, which has important practical significance.

Materials: Great representation of initial materials from this study. Highly recommend to put all used chemicals in this part of Paper, with their origins and purity.

Response:Thanks a lot for your recommendation, we have carefully considered your recommendation and referred to the relevant literature. Among the raw materials, except for the AD, the deionized water is used as a solvent, which has been modified in the paper.In addition, in Section 2.3, a brief introduction to the chemicals

Figures 3 and 4: Scheme of applied experimental equipment are nice represented.

Response:Thank you very much for the encouragement and recognition for this work.